# Some New Aspects of Genetic Variability in Patients with Cutaneous T-Cell Lymphoma

**DOI:** 10.3390/genes13122401

**Published:** 2022-12-18

**Authors:** Vladimír Vašků, Jan Máchal, Filip Zlámal, Anna Vašků

**Affiliations:** 11st Department of Dermatovenereology, St. Anne’s University Hospital, Faculty of Medicine, Masaryk University, 65691 Brno, Czech Republic; 2Department of Pathological Physiology, Faculty of Medicine, Masaryk University, 62500 Brno, Czech Republic

**Keywords:** skin T-cell lymphoma, mycosis fungoides, CTCL, polymorphism-MDR1

## Abstract

Aim: Cutaneous T-cell lymphoma (CTCL) is a group of T-cell malignancies that develop in the skin. Though studied intensively, the etiology and pathogenesis of CTCL remain elusive. This study evaluated the survival of CTCL patients in the 1st Department of Dermatovenereology of St. Anne’s University Hospital Brno. It included analysis of 19 polymorphic gene variants based on their expected involvement in CTCL severity. Material and methods: 75 patients with CTCL, evaluated and treated at the 1st Department of Dermatovenereology of St. Anne´s University Hospital Brno, Faculty of Medicine, Masaryk University, were recruited for the study over the last 28 years (44 men and 31 women, average age 58 years, range 20–82 years). All patients were genotyped for 19 chosen gene polymorphisms by the conventional PCR method with restriction analysis. A multivariate Cox regression model was calculated to reveal genetic polymorphisms and other risk factors for survival. Results: The model identified MDR Ex21 2677 (rs2032582) as a significant genetic factor influencing the survival of the patients, with the T-allele playing a protective role. A multivariate stepwise Cox regression model confirmed the following as significant independent risk factors for overall survival: increased age at admission, clinical staging of the tumor, and male sex. Conclusion: We showed that the TT genotype at position 2677 of the MDR1 gene exhibited statistically significant longer survival in CTCL patients. As such, the TT genotype of MDR1 confers a significant advantage for the CTCL patients who respond to treatment.

## 1. Introduction

Primary cutaneous T-cell lymphoma (CTCL) is a group of T-cell malignancies that develop in the skin [1], and it represents a common group of extra-nodal non-Hodgkin’s lymphoma [2]. The numerous clinical and histological variants of CTCL have been classified by the WHO/EORTEC [1,2]. The most common CTCLs are mycosis fungoides (MF) and Sézary syndrome (SS) [3]. MF is an indolent lymphoma with intermediate aggressiveness. *Sézary syndrome (SES)* is an aggressive form of CTCL derived from central memory T-cells. MF/SS represents approximately 70–75% of all CTCL cases [4].

Patients with CTCL have a highly diverse clinical presentation [5]. In early stages of MF/SS, clinical differential diagnosis can be difficult, with morphology showing large arcuate patches or plaques [6]. Neoplastic cells produce erythema and scaling; with advanced disease, tumors, nodules and/or erythroderma develop [7]. Progression to oncogenesis is poorly understood, but the neoplastic transformation of abnormal T-cells (CD4+ CD45RO+) is probably related to persistent T-cell activation due to antigenic stimulation, increased proliferation, and genomic replication [7].

These malignancies represent proliferation of clonal T-cells, predominantly CD4+ cells that reside in the skin. Skin involvement permits longitudinal assessment, especially skin-directed therapy in early stages, such as topical steroids and phototherapy, to minimize systemic toxicity of drugs [7]. 

Disease dynamics via immunoreaction of the body occurs [8,9,10] with disequilibrium of Th1/Th2 favoring Th2 during MF/SS progression, as defined by changing cytokine profiles. This immune-editing hypothesis distinguishes elimination, equilibrium, and escape phases. In the elimination phase, immunosurveillance occurs, and malignant T-cell clones proliferate. Finally, activation of a Th1 profile occurs. In the equilibrium phase, the malignant T-cells escape and a stabilized state between immunological responses and neoplastic T-cell proliferation develops. Histology shows a heterogeneous infiltrate of neoplastic CD4+ and CD8+ T-cells, which reflects clinical stages I and II. The neoplastic T-cells go through “immune sculpting” in conditions of genomic instability. Tumor antigens become altered, and neoplastic T-cells express immunosuppressive molecules such as IL10, Fas ligand, PD1, and CTLA4 [10,11,12]. The neoplasm grows through its microenvironment into the lymph node, and immunosurveillance is broken [5]. 

Numerous chromosomal abnormalities were detected in patients with MF/SS, which seem to indicate underlying genome instability [7]. Recently, meiosis-specific CT (meiCT) antigens were found to be ectopically expressed in CTCL, which may lead to increased genomic instability by the aberrant process recognized as “meiomitosis.” [13].

Germ cell gene polymorphisms can be associated with CTCL and/or its phenotypes, including pharmacogenetic ones, which are usually reported from cross-sectional studies. FAS (TNF receptor superfamily member 6) participates in T-cell apoptosis and can be deregulated in CTCL. Many CTCL patients are carriers of the homozygous FAS promoter -671 GG SNP, which can blunt its response to interferon [14,15]. An associated genotype of the MMP-2 gene promoter was observed to be more frequent in CTCL-IA stage patients compared with patients with parapsoriasis [16].

Somatic mutations in many genes are common in CTCL, e.g., malignant T-cells with somatic mutations can be detected even in the early stages of MF [2]. Frequent deletions occur in chromatin-modifying genes, in genes responsible for Th2 differentiation, those facilitating escape from growth suppression mediated by TGF-β, and those facilitating resistance to apoptosis mediated by TNFRSF [17]. Changes in remodeling of chromatin, immune surveillance, MAPK signaling, NF-κB signaling, PI-3-kinase signaling, RHOA/cytoskeleton remodeling, and RNA splicing genes also occur in CTCL cells. In summary, CTCL cell somatic mutations are diverse and occur in T-cell signaling, suppressor genes, and cell cycle genes. The putative driver genes remain unrecognized [2]

By transcriptome analysis, 15 genes were implicated in MF progression; all were involved in cell proliferation, immune checkpoints, resistance to apoptosis, and immune responses [18]. 

The genes of interest for this study were selected according to potential etiology/pathogenic contribution to the development of severe skin T-cell lymphoma with limited survival. Those genes participate in systemic inflammation (angiotensinogen—AGT, TNF β, IL-6, adiponectin, leptin, leptin receptor—Lerp), stress (proopiomelanocortin—POMC), vascular wall state and permeability (angiotensin converting enzyme—ACE, endothelin), MMP-mediated proteolysis (matrix metalloproteinase 2-MMP2), and resistance to therapy (multidrug receptor 1-MDR). 

The aim of our research was to perform and evaluate model analysis of survival of a prospective group of CTCL patients including 19 polymorphic variants of candidate genes chosen based on their expected involvement in CTCL pathogenesis/survival. 

## 2. Studied Group

Seventy-five patients with CTCL, evaluated and treated at the 1st Department of Dermatovenereology of St. Anne’s University Hospital Brno, Faculty of Medicine, Masaryk University and at the Centre for treatment of advanced stages of CTCL functioning at the 1st Department of Dermatovenereology, were recruited for the study from 1993–2021. The patient group included 44 men and 31 women, the average age was 58 years, and the range 20–82 years. 

CTCL patients were classified according to Tumor Node Metastasis (TNMB) Classification for Cutaneous T-Cell Lymphoma [19,20,21]. 

According to their stage and clinical state, CTCL patients received topical steroids, a combination of topical steroids with photochemotherapy (PUVA), PUVA alone, photodynamic therapy for residual disease, UVB narrow band therapy, radiotherapy at the Clinic of Radiation Oncology, Masaryk Memorial Institute and Faculty of Medicine, or topical steroids in combination with systemic therapy (interferon α, retinoids, and rexinoids). The chosen treatment led to total initial remission in all patients enrolled in the study. 

All patients were genotyped for 19 chosen polymorphisms by the conventional PCR method with restriction analysis. Peripheral leucocytes were used for DNA analysis, and frozen samples of isolated DNA were stored in a DNA bank at the Department of Pathological Physiology. 

CTCL patients are regularly evaluated, treated, and genotyped for chosen candidate gene polymorphisms. 

This study was approved by the Committee for Ethics of Medical Experiments on Human Subjects, Faculty of Medicine, Masaryk University, Brno (no. 64/93, 1993) and was performed in adherence to the Declaration of Helsinki Guidelines. Participants provided their written informed consent, which has been archived.

## 3. Statistical Analysis

A multivariate Cox regression model was used to evaluate the contribution of genetic polymorphisms and other risk factors to survival. The genetic variants were pre-selected from 19 polymorphisms in candidate genes. The Kaplan–Meier method and log-rank tests in dominant, recessive, and co-dominant mode of allelic expression were used. A *p*-value of 0.1 was used as the cut-off point for including the variable in the analysis. The Hardy–Weinberg equilibrium for each polymorphism was calculated by a Χ^2^ test. Non-genetic variables included in the model were: age at admission, sex, clinical staging (categorized as mild—PP, IA, IB, and/or IIA and severe—IIB, IIIA, and/or IIIB), smoking status, and treatment (local, systemic and/or PUVA).

Supposing an unknown dominance of alleles, three models of gene expression were employed. The genetic factor effect was considered statistically significant only after Bonferroni correction, i.e., *p*-value < 0.017. An all-effect multivariate Cox model was used to determine the hazard ratio of different genotypes of the SNP as a significant factor in overall survival after adjustment for age, sex, staging, smoking status, and treatment.

STATISTICA software (StatSoft, version 12) was used for statistical analysis. MIDAS software (version 1.0) [22] was used for linkage disequilibrium determination.

## 4. Results

All polymorphisms were determined in 75 patients (81% of the initial cohort of 93 patients), and only those subjects were included in subsequent analyses. Basic characteristics of the group are listed in Appendix A.

The five-year survival of the cohort was 86%; the median survival was 229 months (19 years). The Kaplan–Meier curve is shown in Appendix A.

Minor allele frequencies of all polymorphisms are presented in Appendix A. The calculated regression model identified MDR Ex21 2677 (rs2032582) as a significant genetic factor influencing the overall survival of the patients, with the T-allele playing a protective role. TNFb NcoI (rs909253) showed a similar trend, but it was not significant at the pre-specified level of α = 0.017. The hazard ratios and *p*-values in different models are listed in Appendix A. From other factors, age, sex, and clinical staging were identified as significant prognostic factors in all models.

The Kaplan–Meier survival analysis for different genotypes of MDR Ex21 2677 are shown in Appendix A. The effect of MDR Ex21 2677 on the overall survival when adjusted for age, sex, staging, smoking status, and treatment is shown in Appendix A (GG is a reference genotype). The TT genotype was associated with longer survival (*p* = 0.027).

Considering all clinical features, a multivariate stepwise Cox regression model confirmed the following as significant independent risk factors for overall survival: increased age at admission, clinical staging of the tumor, and male sex (Appendix A).

Given the data, MDR Ex21 2677 (rs2032582) is in linkage equilibrium with MDR Ex26 3435 C/T (rs1045642), which is situated 758 bp downstream in the same gene. This linkage could affect eventual associations of MDR Ex26 3435 (rs1045642) with the survival (not confirmed in our study). The analysis revealed significant linkage disequilibrium, with haplotypes G-C and T-T significantly more abundant compared to linkage equilibrium (D’ = 0.80, *p* < 10^−6^ for the G-C haplotype; D’ = 0.79, *p* < 10^−6^ for the T-T haplotype). The T-T haplotype was the most abundant (41%) in our study population. G-C was present in 37% of patients, G-T in 15% of patients, T-C in 4% of patients, A-C in 2% of patients, and A-T was absent in patients.

## 5. Discussion

We found that 86% of CTCL patients evaluated and treated in Brno survive more than five years from initial diagnosis of CTCL, with a median survival of 19 years. Four independent risk factors for survival were identified (age at diagnosis, sex, and clinical staging) including one of 19 evaluated polymorphisms: MDR1.

The transporter P-glycoprotein coded by the MDR1/ABCB1 gene at 7q21 is expressed in epithelial cells in many tissues and organs including skin [23,24].

The MDR1 gene is composed of two promoters and 28 exons [25]. A Major downstream/proximal promoter regulates most transcriptional activity of the gene using three CpG islands. MDR1 transcription can be modified by methylation of the three CpG islands as well as by histone modifications [25,26]. 

P-glycoprotein is a membrane protein and acts as an ATP-binding cassette (ABC) transporter that provides protection against xenobiotics [27,28,29,30,31]. 

The putative role of the MDR1 in human skin is unknown. High expression of MDR1 mRNA was detected in the human dermis but not in the epidermis. MDR1 expression in the dermis is not associated with fibroblasts. MDR1 expression in SCC cells was lower compared with expression in normal skin. High production of P-gp protein was also observed in sweat ducts, vessels, nerve sheaths and muscles of human skin. Dermal vasculature seems to represent a barrier for xenobiotic uptake to skin [32,33]. Drug delivery and transport was also associated with high expression of ABCB1 in the skin and its appendages, which could be associated with therapeutic responsiveness to various agents, including topical steroids, methotrexate, cyclosporine, azathioprine, antihistamines, antifungal agents, colchicine, tacrolimus, ivermectin, tetracycline, retinoid acids, and biologic agents [34]. Genetic variability in the MDR1 gene was associated with the pathogenesis of several dermatoses, including psoriasis, atopic dermatitis, melanoma, bullous pemphigoid, Behçet disease, and lichen planus [34]. 

The MDR1 transporter also affects individual susceptibility to viral infection and to complex diseases [35,36,37]. 

The MDR1 transporter has been found to modify pharmacokinetic effects of many frequently used drugs including anti-cancer drugs [38]. 

Transformed and non-transformed cell types increase transcription of MDR1 in hypoxic conditions; the MDR1 promoter contains a functional HIF-1 binding site [39]. Higher expression of P-gp during hypoxia may explain resistance of some tumors to chemotherapy [40].

Three single nucleotide polymorphisms (C3435T, C1236T, and G2677T/A) in the MDR1 gene are frequently studied. The C-G-T haplotype constructed from their alleles in linkage disequilibrium was associated with an increased risk of T-cell lymphoma [41]. Genotypes and allelic as well as haplotype associations with many chronic inflammatory diseases and cancers were found [42], but the G2677T/A polymorphism showed controversial results with increased, decreased, or no functional effect of the T-allele [42,43,44,45,46], which is not surprising when age at diagnosis and staging of CTCL are included in the survival model for CTCL [47,48]. The male sex risk deserves explanation since the TT genotype was associated with twice the male infertility, suggesting possible detrimental effects of environmental factors including working conditions faced by men [49]. We showed that the TT genotype at position 2677 of the MDR1 gene was associated with statistically significant longer survival in CTCL patients. We also showed that the TT genotype of MDR1 is advantageous for CTCL patients responsive to a repertoire of clinically successful treatments [50]. 

Until now, the pathogenetic pathways responsible for CTCL manifestation are not fully understood. Thus, it is difficult to formulate a truly a priori candidate gene hypothesis. However, a hypothesis-driven approach reflecting candidate gene studies may offer a reasonable approach for understanding the germ cell gene predisposition for CTCL. associated with CTCL. We found significant associations of polymorphisms in the MDR1 gene in genotype–phenotype studies together with other independent risk factors for CTCL. The TT genotype was strongly associated with longer survival in CTCL patients. We plan to use these results as basis for further study to evaluate this genetic association with respect to developing time dynamics of CTCL in affected patients.

## Data Availability

The data presented in this study are available on request from the corresponding author.

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
