# Peer review of "Some New Aspects of Genetic Variability in Patients with Cutaneous T-Cell Lymphoma"

_genes, 2022, doi:10.3390/genes13122401_

Round 1

Reviewer 1 Report

In this manuscript, the authors evaluated 19 polymorphic variants of genes in patients with CTCL, and analyzed the effects of these variants on survival. Although the authors identified a significant genetic factor MDR1 in CTCL, the manuscript is poorly written in the scientific and linguistic point of view; the logic and rationale of this study are chaotic and obscured.

1. The Introduction section is illogical and chaotic. For example, this study is mainly focused on the genetic variability of CTCL, but the authors introduced largely about the immunomarkers and microRNAs in terms of CTCL. Little is introduced about the recent discoveries on the genetic abnormalities in CTCL, as well as the significance of this study.

2. Many genetic abnormalities have so far been reported in MF/SS. How and why did the authors selectively choose 19 polymorphic variants of genes? In this regard, the rationale of this study is unclear. The reasons and supporting references should be provided in either the Introduction section or the Result section, but not the Methods section.

3. The manuscript has so many grammatical, tense and format errors, as well as punctuation mistakes. A thorough English editing is warranted.

Author Response

Thank you for the review.

According to your review, we substantially re-write the article with respect to scientific content as well as to the linguistic level.

  1. Introduction as well as Discussion are re-written. Immunomarkers information was limited; information about genetic abnormalities in CTCL patients was enriched.
  2. The choice of germ cell genetic polymorphisms is explained in Introduction. We preferred hypothesis- based association study respecting etiology/ pathogenic aspects of CTCL. Genome-wide study approach will bring larger information, but interpretation of results is so far difficult, especially in rare diseases as CTCL.
  3. After our scientific revision of manuscript, English revision was performed by American native speaker –experienced and respected dermatologist.   

Reviewer 2 Report

I congratulate the authors for the long term study.

Still I do not understand if this is a prospective study, started in 1993, or a retrospective one.

Genomic testing was performed from what kind of biologic material? How was it preserved since 1993?

In the end of the article, the conclusions are not so clear. Which is the purpose of the study and what are the benefits for the patients?

Author Response

Thank you very much for the review.

Genetic testing was performed from our stored samples of isolated DNA from peripheral leucocytes, because we focused on germ-cell polymorphisms; genotyping was realized gradually. Genotyping is organizing as cross-sectional association study of genotype-phenotype. Clinically, the study was scheduled as prospective, survival as a parameter of severity was used.    

Introduction as well as Discussion are re-written. We tried to explain better the purpose of the study as well as benefit for patients.

Round 2

Reviewer 1 Report

The manuscript is much improved and most of the comments have been addressed except for some spelling errors.